# Missing Heritability in Albinism: Deep Characterization of a Hungarian Albinism Cohort Raises the Possibility of the Digenic Genetic Background of the Disease

**DOI:** 10.3390/ijms25021271

**Published:** 2024-01-20

**Authors:** Nikoletta Nagy, Margit Pal, Jozsef Kun, Bence Galik, Peter Urban, Marta Medvecz, Beata Fabos, Alexandra Neller, Aliasgari Abdolreza, Judit Danis, Viktoria Szabo, Zhuo Yang, Stefanie Fenske, Martin Biel, Attila Gyenesei, Eva Adam, Marta Szell

**Affiliations:** 1Department of Medical Genetics, University of Szeged, 6720 Szeged, Hungary; pal.margit@med.u-szeged.hu (M.P.); alexa.neller@gmail.com (A.N.); aliasgari.abdolreza@gmail.com (A.A.); dr.eva.adam@gmail.com (E.A.); szell.marta@med.u-szeged.hu (M.S.); 2HUN-REN-SZTE Functional Clinical Genetic Research Group, 6720 Szeged, Hungary; 3Hungarian Centre for Genomics and Bioinformatics, Szentagothai Research Centre, University of Pecs, 7624 Pecs, Hungary; kun.jozsef@pte.hu (J.K.); galik.bence@pte.hu (B.G.); urban.peter@pte.hu (P.U.); gyenesei.attila@pte.hu (A.G.); 4Department of Dermatology, Venereology and Dermatooncology, Semmelweis University, 1095 Budapest, Hungary; medvecz.marta@med.semmelweis-univ.hu; 5ERN-Skin Reference Centre, Semmelweis University, 1095 Budapest, Hungary; 6Mor Kaposi Teaching Hospital of Somogy County, 7400 Kaposvar, Hungary; fabosbeata@gmail.com; 7HUN-REN-SZTE Dermatological Research Group, 6720 Szeged, Hungary; danis.judit@med.u-szeged.hu; 8Department of Immunology, University of Szeged, 6720 Szeged, Hungary; 9Department of Ophthalmology, Semmelweis University, 1085 Budapest, Hungary; 10Department of Pharmacy, Center for Drug Research, Ludwig-Maximilians-Universität München, 81377 Munich, Germanymbiel@cup.uni-muenchen.de (M.B.)

**Keywords:** albinism, missing heritability, digenic inheritance, *TPCN2* gene

## Abstract

Albinism is characterized by a variable degree of hypopigmentation affecting the skin and the hair, and causing ophthalmologic abnormalities. Its oculocutaneous, ocular and syndromic forms follow an autosomal or X-linked recessive mode of inheritance, and 22 disease-causing genes are implicated in their development. Our aim was to clarify the genetic background of a Hungarian albinism cohort. Using a 22-gene albinism panel, the genetic background of 11 of the 17 Hungarian patients was elucidated. In patients with unidentified genetic backgrounds (*n* = 6), whole exome sequencing was performed. Our investigations revealed a novel, previously unreported rare variant (N687S) of the two-pore channel two gene (*TPCN2*). The N687S variant of the encoded TPC2 protein is carried by a 15-year-old Hungarian male albinism patient and his clinically unaffected mother. Our segregational analysis and in vitro functional experiments suggest that the detected novel rare *TPCN2* variant alone is not a disease-causing variant in albinism. Deep genetic analyses of the family revealed that the patient also carries a phenotype-modifying R305W variant of the OCA2 protein, and he is the only family member harboring this genotype. Our results raise the possibility that this digenic combination might contribute to the observed differences between the patient and the mother, and found the genetic background of the disease in his case.

## 1. Introduction

Albinism is characterized by a variable degree of hypopigmentation of the skin and the hair, and ophthalmologic symptoms including nystagmus, foveal hypoplasia, chiasmatic misrouting of the optic nerves, iris transillumination, retinal hypopigmentation and reduced visual acuity [1,2,3]. Non-syndromic oculocutaneous, ocular and syndromic forms of albinism are inherited in an autosomal or X-linked recessive manner and, so far, are associated with 22 disease-causing genes [4,5,6,7,8,9]. Screening for these known causative genes has an approximately 70% diagnostic yield for albinism, as exemplified by a gene-panel study of a cohort of 990 patients that elucidated the genetic background for 72.3% of the patients [10]. This example well demonstrates that missing heritability affects approximately one quarter of albinism patients [9]. Missing heritability shows high variability in rare monogenic diseases: it can be extremely low, as, for example, in the skin tumor syndrome, CYLD cutaneous syndrome, and it can be relatively high, as it is for albinism [10,11].

The problem of missing heritability can be—at least in part—explained by a yet undiscovered mode of inheritance. Recently, a case with dominant inheritance of albinism was published. The first reported case of dominant albinism is caused by a newly discovered mutation (R210C) of the two-pore channel two (*TPCN2*) gene, recently identified in a Chinese child by Wang et al. [12].

The *TPCN2* gene is located in 11q13.3 (HGNC:20820) and encodes a protein of 752 amino acids. The encoded TPC2 channel protein contains two homologous repeats of six transmembrane domains (S1–S6) that form two shaker-like domains (I and II), each comprising a voltage sensor (S4) and an ion conducting pore (S5–S6) [13,14]. Two TPC2 proteins form a homodimer displaying a pseudo-fourfold symmetry, thus forming a functional non-selective cation channel [13,14]. TPC2 can be activated by phosphatidylinositol 3,5-bisphosphate (PI(3,5)P2) or by nicotinic acid adenine dinucleotide phosphate (NAADP) binding [15]. TPC2 regulates melanosome pH and membrane potential and, consequently, pigment production [12,13,14].

In this paper, we report on a genetic investigation of a 17-member Hungarian albinism cohort. The results of our 22-gene panel survey were in agreement with the worldwide accepted genetic yield (apr. 70%) for albinism. Furthermore, the analyses of the WES results of an affected family raise the possibility of the digenic nature of the disease.

## 2. Results

For genetic screening of the Hungarian albinism cohort (*n* = 17), a panel of 22 known disease-causing genes with autosomal or X-linked recessive inheritance was used, and the genetic backgrounds of 11 patients, 64.7% of the cohort, were identified (Table 1). Regarding these 11 patients, genetic investigations revealed that 10 of them carry pathogenic genetic variants in tyrosinase genes (*TYR*), solute carrier family 45 member 2 (*SLC45A2*) and oculocutaneous albinism type 2 (*OCA2*), responsible for the development of the non-syndromic forms of albinism, and one of the 11 patients carries pathogenic variants of a gene (biogenesis of lysosomal organelles complex 3 subunit 1 (*HPS1*) gene) associated with the syndromic form of the disease. Seven patients are compound heterozygotes for two pathogenic variants of the *TYR* gene and belong to the oculocutaneous albinism (OCA) type 1 (OCA1) subtype of the disease. Two patients carry two heterozygous pathogenic variants of the *SLC45A2* gene and belong to the OCA type 4 (OCA4) group of the disease. In one patient, heterozygous pathogenic variants of the *OCA2* gene are present, indicating the OCA type 2 (OCA2) subgroup. In one patient, genetic investigations revealed a syndromic form of the disease. This patient is a carrier of heterozygous pathogenic variants of the *HPS1* gene confirming Hermansky–Pudlak syndrome-1 (HPS1) (Table 1). All identified pathogenic variants have been reported previously [4,5,6,7,8,9].

WES of the cases for which the genetic background was not identified (*n* = 6) revealed one novel rare missense variant (c.2060A>G p.N687S) of the *TPCN2* (NM_139075.4) gene. This gene has recently been implicated (2023) in the newly discovered dominant type of albinism [12]. The N687S variant is present in a 15-year-old Hungarian male patient (No. 16) in heterozygous form. Clinical characteristics of the patient are summarized in Appendix A. The variant was confirmed with Sanger sequencing (Figure 1A). The clinically unaffected mother of the patient also carries the N678S variant in heterozygous form. Neither the father nor the tested 87 Hungarian controls carry this variant.

The clinical consequences of the novel N687S variant are expected to be pathogenic: “deleterious” by the SIFT algorithm, “probably damaging” by the PolyPhen tool, “likely disease-causing” by the REVEL method, “damaging” by the MetaLR method and “medium” by the Mutation Assessor analysis tools of Ensemble Genome Browser, http://www.ensembl.org (accessed on 13 January 2024). According to the Ensemble Genome Browser, the frequency of the rare allele (G) of the N687S variant (rs150476703) is 0.0001999. The N687S variant is within the sixth transmembrane region of the second shaker-like domain containing the second of six transmembrane regions of the TPC2 protein (SWISS-MODEL Repository–Expasy, https://swissmodel.expasy.org (accessed on 13 January 2024); Figure 1B). The affected amino acid residue is in an evolutionary conserved region of the protein (Figure 1C).

We have performed an analysis of the common *TPCN2* variants—some of which have already been implicated as modifiers of pigmentation [16,17]—in the WES analyzed Hungarian albinism cohort (*n* = 6) and in controls (*n* = 87) (Appendix A). No significant difference was observed between patients and controls for the allele frequencies of any of the detected frequent missense variants of the *TPCN2* gene in this Hungarian cohort. The detected common missense *TPCN2* variants are not linked.

To investigate the putative functional effect of the N687S TPC2 variant, we performed endolysosomal patch-clamp experiments. No difference was detected in the basal channel activity, or in the channel activity after stimulation with PI(3,5)P2, between the novel rare N687S variant and the WT protein. We also compared the channel activity of the M484L variant with a known gain-of-function effect and the double polymorphic M484L/N687S variant, and found no differences in their basal channel activity or in the channel activity after stimulation with PI(3,5)P2 (Figure 2).

There is increasing evidence that common variants in different albinism genes can have an additive effect and modify a phenotype [18]. Therefore, besides the polymorphisms of the *TPCN2* gene, common variants of known pathogenic albinism genes were analyzed. In the total study population, four common variants of known albinism genes were detected with high frequencies: the L374F variant of the SLC45A2 protein, the R305W of the OCA2 protein and the R402Q and S192Y TYR proteins (Table 1).

WES analyses of patient No. 16 and his parents have also been performed. Screening the results of WES analyses for the presence of the same common phenotype modifying polymorphisms revealed the R305W variant of the OCA2 protein in the patient and in his father, but not in his mother (Table 2). The co-occurrence of the novel N687S TPC2 and R305W OCA2 variants (highlighted with bold) was detected only in patient No. 16, but not in either of his parents.

## 3. Discussion

Here we report that the genetic screening of 17 Hungarian albinism patients was performed with a panel of 22 established albinism-causing genes. The genetic background of the disorder was identified for 11 patients (64.7% diagnostic yield), which corresponds to missing heritability of 35.3% in this Hungarian cohort.

Besides identifying the pathogenic variant in the 11 albinism patients, the panel sequencing also made it possible to analyze the common variants of the entire cohort: a recent study suggested that *TYR* variants S192Y and R402Q are likely to be significant modifiers of other pigmentation gene variants. The double-variant 192Y-402Q haplotype occurs at a low frequency and is likely to be deleterious [18]. In our patient cohort, the two variants were present together in three of 17 albinism patients (Patients No. 3, No. 9 and No. 10 in Table 1) but genotype–phenotype analysis could not identify any significant effect of the double-variant on the phenotype of the patients.

For the six cases for which no known disease-causing variants were detected, we performed WES. Our WES analysis in these Hungarian albinism patients with unelucidated genetic backgrounds identified a novel rare heterozygous variant N687S of the TPC2 protein in patient No. 16. None of the pathogenic or likely pathogenic variants of the 22 known disease-causing genes associated with albinism were present in this patient.

The N687S novel rare variant, first reported in this paper, was found in a 15-year-old Hungarian male patient with hypopigmented skin, blue eye color, light blond,-brown hair and decreased foveal reflex. The Hungarian patient is the only family member affected by albinism, although his father and grandfather both have blond hair and blue eyes, and other relatives have red hair. Besides the patient, the N687S variant is only carried by his symptom-free mother, not by his father. The unrelated Hungarian controls (*n* = 87) included in this study did not carry the N687S TPC2 variant.

Results of the prediction analysis revealed the mutation to be deleterious, and to verify this we performed functional studies. The detected N687S pathogenic variant is in the middle of the IIS6 region of the TPC2 protein. The IIS6 region is linked with the motion of the C-terminal end of the TPC2 protein and is involved in PI(3,5)P2-induced channel opening [12,13,14,15,16,17,18,19]. The R210C variant, reported in the affected Chinese patient, is located in the IS4-S5 linker helix of the protein, and electrophysiological examination demonstrated that this variant causes constitutive channel activation and increased affinity to PI(3,5)P2 [11]. Our endolysosomal patch-clamp experiments showed that the novel rare N687S variant has no effect on basal or on PI(3,5)P2-stimulated channel activity, excluding a pathomechanistic effect of this rare variant. However, we cannot exclude that, with other yet unknown mechanisms, this variant might contribute to pigmentation and thus to the phenotypic diversity in albinism.

In the WES examined six patients, in whom the 22-gene panel could not detect any pathogenic variants, the co-occurrence of the TYR R402Q and S192Y common polymorphisms could not be detected (Patients No. 12, 13, 14, 15, 16 and 17 in Table 1). To note, the R305W variant of *OCA2*, which is associated with eye and skin color, was detected in eight of 17 patients, while the L374F variant, which affects pigmentation, was present in all albinism patients studied. Patient No. 16 with the N687S TPC2 variant carries the L374F variant of the *SCL45A2* gene in homozygous form, the R402Q variant of the *TYR* gene and the R305W variant of the *OCA2* gene in heterozygous form (Patient No. 16. in Table 1). Among these, only the R305W OCA2 variant is not present in the unaffected mother (Table 2). The *OCA2* gene encodes a protein, which contributes to a melanosome-specific anion (chloride) current that modulates melanosomal pH for optimal tyrosinase activity required for melanogenesis and the melanosome maturation [20].

In this study, we investigated a Hungarian albinism cohort for the genetic background of the disease. (1.) We found that in 35.3% of the cohort (*n* = 6), albinism could not be explained by pathogenic mutations of established albinism genes. (2.) WES analyses of these six patients revealed a novel rare TPC2 N687S variant in heterozygous form in patient No. 16. (3.) WES analyses of the parents showed that only the phenotypically unaffected mother carries this variant, also in heterozygous form. (4.) In vitro endolysosomal patch-clamp experiments did not show any effect of the novel rare N687S variant on basal or on PI(3,5)P2-stimulated channel activity. (5.) Analyses of common polymorphisms of known albinism genes revealed the presence of the phenotype-modifying R305W OCA2 variant in patient No. 16, but not in his mother.

The results of our survey demonstrate that the detected novel N687S TPC2 variant alone cannot be regarded pathogenic in albinism; however, we cannot exclude the possibility that in combination with trans variants such as the R305W of the OCA2 protein, it could influence pigmentation. Both OCA2 and TPC2 are melanosomal expressed channel proteins, and they are key in the regulation of melanosomal pH and consequently in pigment production. On these grounds, it can be assumed that the double mutant genotype might modify or aggravate the phenotypes of the patients, and thus contribute to the relatively high phenotypic diversity seen in albinism. There are several well documented examples in the literature of digenic inheritance, when heterozygous mutation in each of two genes is not sufficient to produce a recognizable phenotype but their co-existence is necessary for the development of the disease [21]. The explanation for missing heritability in albinism—besides the recently described dominant inheritance—might also be hidden in digenic inheritance.

## 4. Materials and Methods

### 4.1. Patients

Seventeen Hungarian patients with albinism and 87 Hungarian individuals without albinism (controls) were enrolled in the study. Patients were diagnosed with the French guidelines for albinism [22]. Written informed consent was obtained from all the enrolled individuals according to a protocol approved by the Hungarian National Public Health Centre in adherence to the Helsinki guidelines. DNA was extracted from peripheral blood using QIAGEN kits.

### 4.2. The 22-Gene Panel for Albinism Genetics

Our gene panel contained the following 22 genes: *TYR*, *OCA2*, *TYRP1*, *SLC45A2*, *SLC24A5*, *LRMDA*, *DCT*, *PMEL*, *GPR143*, *SLC38A8*, *HPS1*, *AP3B1*, *HPS3*, *HPS4*, *HPS5*, *HPS6*, *DTNBP1*, *BLOC1S3*, *BLOC1S6*, *AP3D1*, *BLOC1S5*, *LYST*.

### 4.3. Whole-Exome Sequencing (WES)

Genotypes of patients were determined using a targeted next-generation sequencing (NGS) approach. Libraries were prepared using the DNA Prep with Enrichment and Illumina Exome Panel probes (Illumina, Inc., San Diego, CA, USA). Pooled libraries were sequenced on the Illumina NovaSeq 6000 NGS platform (Illumina, Inc., San Diego, CA, USA). Adapter-trimmed and Q30-filtered paired-end reads were aligned to the hg19 Human Reference Genome using the Burrows–Wheeler Aligner (BWA, Illumina, San Diego, CA, USA). Duplicates were marked using the Picard software package version 2.22.1. The Genome Analysis Toolkit (GATK) was used for variant calling (BaseSpace BWA Enrichment Workflow v2.1.1. with BWA 0.7.7-isis-1.0.0, Picard: 1.79 and GATK v1.6-23-gf0210b3).

As a result of sequencing, the mean on-target coverage was 71× per base with an average percentage of targets covered greater than or equal to 30× of 96% and 90%, respectively. Variants passed by the GATK filter were used for downstream analysis and annotated using the ANNOVAR software tool (version 17 July 2017). Single-nucleotide-polymorphism testing was performed as follows: high-quality sequences were aligned with the human reference genome (GRCh37/hg19) to detect sequence variants, and the detected variations were analyzed and annotated. Variants were filtered according to read depth, allele frequency and prevalence reported in genomic variant databases, such as ExAc (v.0.3) and Kaviar. Variant prioritization tools (PolyPhen-2, SIFT, LRT, Mutation Assessor) were used to predict the functional impact of the mutation. For variant filtering and interpretation, VarSome and Franklin bioinformatic platforms (https://franklin.genoox.com, accessed on 13 January 2024) were used, incorporating the guidelines of the American College of Medical Genetics and Genomics [23].

All the identified candidate variants were confirmed by bidirectional capillary Sanger sequencing.

### 4.4. Chimaeric Constructs

The YFP-tagged TPC2(L564P) pcDNA3.1 plasmid DNA was a kind gift of Dr. Martin Biel. In the text, this construct is referred to as wt. TPC2(M484L) and TPC2(N687S) mutations were generated in the YFP-tagged TPC2(L564P) plasmid DNA using the Quick Change Site-Directed Mutagenesis Kit (Agilent, Santa Clara, CA, USA) according to the instructions of the manufacturer. Polymorphic TPC2 variants were also generated in the TPC(564P)-YFP background. The final constructs carrying mutant cDNAs were verified by Sanger sequencing. To evaluate expression levels of TPC2 WT and variants in endolysosomal membranes, HEK293 cells were transfected with YFP-tagged TPC2-constructs. Preparation of endolysosomes was performed as described previously [17].

### 4.5. Whole-Endolysosomal Patch-Clamp Experiments

Human TPC2 WT and polymorphic TPC2 variants (C-terminally fused to YFP) were transiently transfected into HEK293 cells using TurboFect Transfection Reagent (Thermo Fisher, Waltham, MA, USA). For whole-endolysosomal patch-clamp recordings, isolated intact endolysosomes from HEK293 cells were manually isolated after vacuolin treatment for at least 2 h. Currents were recorded using an EPC-10 patch-clamp amplifier and PatchMaster acquisition software (HEKA, www.heka.com, accessed on 13 January 2024). Data were digitized at 40 kHz and filtered at 2.8 kHz. Cytoplasmic solution contained 140 mM potassium methanesulfonate (KMSA), 5 mM KOH, 4 mM NaCl, 0.39 mM CaCl_2_, 1 mM EGTA, and 20 mM HEPES (pH adjusted with KOH to 7.2). Luminal solution was 140 mM NaMSA, 5 mM KMSA, 2 mM CaMSA, 1 mM CaCl_2_, 10 mM HEPES, and 10 mM MES (pH adjusted with MSA to 4.6). For measurements with PI(3,5)P2 (water-soluble diC8 form, from Echelon Biosciences, Salt Lake City, UT, USA), PI(3,5)P2 was added to the cytoplasmic solution to give a final concentration of 1µM. 500 ms voltage ramps from −100 to +100 mV were applied every 5 s from a holding potential of 0 mV. Current amplitudes at −100 mV were determined from individual ramp current recordings, and current density was calculated by dividing by cell capacitance. All recordings were performed at 23–25 °C.

### 4.6. Statistical Analysis

Statistical analysis was carried out using a one-way ANOVA test, followed by Tukey’s post-hoc test, and was applied to basal and PI(3,5)P2 conditions.

## 5. Conclusions

The well-known forms of albinism are inherited in an autosomal or X-linked recessive manner and are associated with 22 disease-causing genes. The *TPCN2* gene has been recently implicated (2023) as a novel disease-causing gene in the newly discovered dominant type of albinism. Here we report on the detection of a novel, rare TPC2 variant N687S in an albinism patient in heterozygous form. Deep genetic analyses of the family revealed that the patient carries a phenotype-modifying variant of the *OCA2* gene, and he is the only family member harboring this digenic combination. Our findings draw attention to the possible digenic genetic background of albinism. On this ground we recommend re-analysis—possibly utilizing newly developed mathematical models [24,25]—of already existing genetic databases of albinism cohorts to fill the gap of missing heritability of the disease.

## Figures and Tables

**Figure 1 ijms-25-01271-f001:**
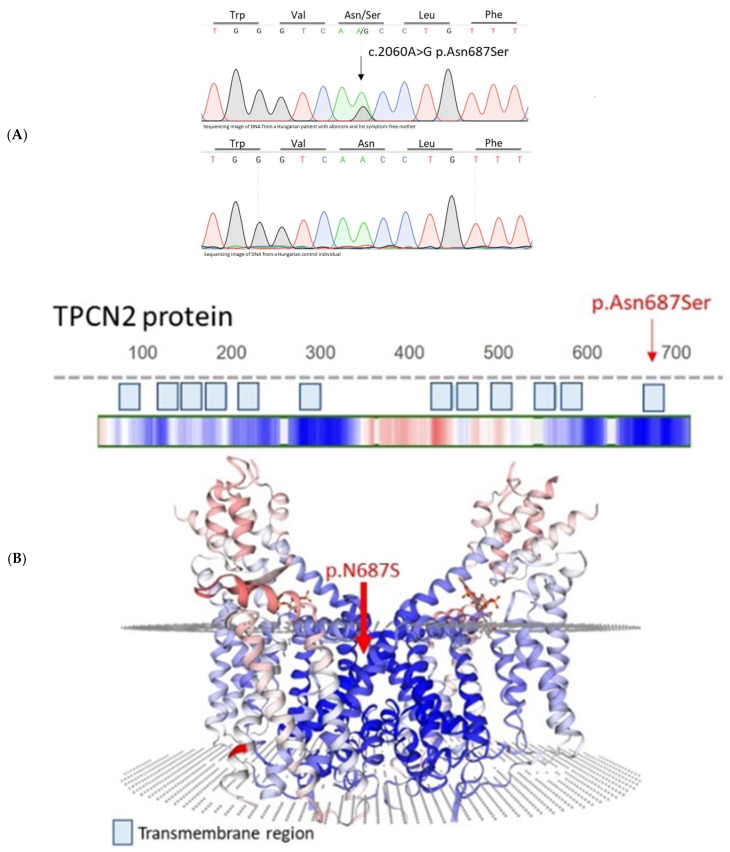
Characterization of the novel N687S TPC2 variant. (**A**): Sanger sequencing result of the novel rare N687S TPC2 variant. The four colors represent the four bases of the DNA: guanine (black), cytosine (blue), thymine (red) and adenine (green). (**B**): The N687S variant is present in the sixth transmembrane region of the second shaker-like domain containing the second of six transmembrane regions of the TPC2 protein (SWISS-MODEL Repository–Expasy, https://swissmodel.expasy.org, accessed on 13 January 2024). (**C**): Evolutionary conservation of the region of the N687S variant (AMINODE evolutionary analysis, www.aminode.org, accessed on 13 January 2024). The yellow bar indicates evolutionary conserved region. The red bar indicates the relative substitution score.

**Figure 2 ijms-25-01271-f002:**
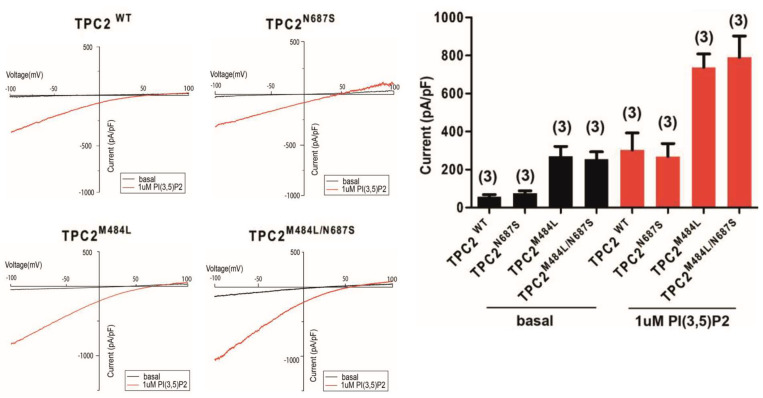
Representative current-voltage relationships recorded from vacuolin-enlarged endolysosomal vesicles of HEK293 cells expressing TPC2 WT, TPC2N687S, TPC2M484L or TPC2M484L/N687S. Currents were obtained before (basal, black) and after application of 1 μM phosphatidylinositol 3,5−bisphosphate (PI(3,5)P2) (red) (**left**). Statistical analysis of current densities calculated from currents determined at −100 mV as shown on the left. The number of experiments is indicated in brackets. Error bars indicate SEM (**right**).

**Table 1 ijms-25-01271-t001:** Screening results from a panel of 22 disease-causing genes and common phenotype modifying variants for 17 Hungarian patients with albinism. ND indicates not detected.

	Pathogenic Variants				Common Polymorphisms			
Patients	Mutation 1	Mutation 2	Gene	Type	L374F *SLC45A2*	S192Y *TYR*	R402Q *TYR*	R305W *OCA2*
1	p.Gln487Ter	p.Gly409Asp	*SLC45A2*	OCA4	homozygous			
2	p.Val367Ile	p.Gly198Asp	*SLC45A2*	OCA4	homozygous	heterozygous		
3	p.Asn489Asp	p.Val443Ile	*OCA2*	OCA2	heterozygous	heterozygous	heterozygous	heterozygous
4	p.Arg217Gln	p.Ala490fs	*TYR*	OCA1	homozygous		heterozygous	
5	p.Arg217Gln	p.Ala490fs	*TYR*	OCA1	homozygous		heterozygous	
6	p.Val183Leu	p.Arg402Ter	*TYR*	OCA1	homozygous	heterozygous		
7	p.Pro21Ser	p.Met96fs	*TYR*	OCA1	heterozygous			heterozygous
8	p.Pro21Ser	p.Met96fs	*TYR*	OCA1	heterozygous			heterozygous
9	p.Arg217Gln	p.Arg402Ter	*TYR*	OCA1	homozygous	heterozygous	heterozygous	heterozygous
10	p.Pro21Ser	p.Val183Leu	*TYR*	OCA1	homozygous	homozygous	heterozygous	
11	p.Tyr245Leufs	P.Gly321Alafs	*HSP1*	HSP1	homozygous	heterozygous		
12	ND	ND	ND	ND	homozygous	homozygous		heterozygous
13	ND	ND	ND	ND	homozygous			heterozygous
14	ND	ND	ND	ND	homozygous	heterozygous		
15	ND	ND	ND	ND	homozygous	heterozygous		
16 *	ND	ND	ND	ND	homozygous		heterozygous	heterozygous
17	ND	ND	ND	ND	homozygous	homozygous		heterozygous

* Patient No. 16 carries the novel N687S TPC2 variant.

**Table 2 ijms-25-01271-t002:** Presence of the novel N687S and the common phenotype modifying *TPCN2* variants in Patient No. 16 and his parents.

Variant	N687S *TPCN2*	S192Y *TYR*	R402Q *TYR*	R305W *OCA2*	L374F *SLC45A2*
Patient No. 16	heterozygous	WT	heterozygous	heterozygous	homozygous
Mother of the patient	heterozygous	WT	heterozygous	WT	homozygous
Father of the patient	WT	heterozygous	WT	heterozygous	homozygous

## Data Availability

The data presented in this study are available on request from the corresponding author. The data are not publicly available because they are genetic data.

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
