# Peer review of "Missing Heritability in Albinism: Deep Characterization of a Hungarian Albinism Cohort Raises the Possibility of the Digenic Genetic Background of the Disease"

_ijms, 2024, doi:10.3390/ijms25021271_

Round 1

Reviewer 1 Report (New Reviewer)

Comments and Suggestions for Authors

The authors describe a genetic investigation using a 22-gene panel for Albinism genetics, of a 17-member Hungarian albinism cohort. 11 patients have a molecular confirmation of the diagnosis.

In one patient, they identified one variant of unknown significance in TPC2, inherited from healthy mother. In addition, functionnal studies on this variant suggest it has no impact of the channel activity.In this situation, TPCN2 may have nothing to do with albinism in this patient, and the title of the paper seems not to be appropriate.

-It could have been of interest to get more details regarding the phenotype of the patients, in particular those without any confirmation of the diagnosis.

-Has the WES been performed in trio ? much more powerful in this situation.

Author Response

Reviewer 2 Report (New Reviewer)

Comments and Suggestions for Authors

The authors identified a novel mutation in TPCN2 that was present in a 15 year old albino Hungarian. The only closest relative that also presents the mutation is the mother but is asymptomatic. Though no direct test can confirm that the mutation is causative for the Albinism in the 15 year old patient, in absence of other known mutations the authors believe that the point mutation plays a significant role in disease presentation. 

1. The author have identified the same point mutation in other individuals that present no sign of Albinism. Have the author an explanation for this?

2. The authors have analyzed a frequency of other point mutations that occur along with the identified mutation. Is there a combination of point mutation that is present in the patient but absent in the mother? Possibly X-linked?

There is a grammatical error in the introduction-

(TPCN2) gene, recently identified in a Chinese child by Wang et al. in a Chinese child. 

Comments on the Quality of English Language

Common grammatical errors that require correction.

Author Response

This manuscript is a resubmission of an earlier submission. The following is a list of the peer review reports and author responses from that submission.

Round 1

Reviewer 1 Report

Comments and Suggestions for Authors

The authors present the analysis of patients with albinism. Analysis with a specific gene panel established the molecular diagnosis in 11/17 patients.

In patients for whom the molecular diagnosis was not obtained, they performed WES. In 1 patient they identified a heterozygous variant in the TPCN2 gene,  NM_139075.4:c.2060A>G; p.N687S. This variant is a VUS, and is inherited from his healthy mother. It therefore does not explain the phenotype of the patient.

There are several concerns with this manuscript.

1) The title puts the emphasis on a variant that is a VUS and has no clinical impact on the patients phenotype. This is misleading.

2) The manuscript first describes analysis of a cohort of 17 patients with an albinism panel, showing a diagnosis in 11 patients. Then for those in whom the diagnosis was not achieved, the authors performed WES, which led to identification of the TPCN2 variant, NM_139075.4:c.2060A>G; p.N687S. This variant is a VUS, that is not responsible for the patient's phenotype. The authors then performed an analysis of various already described TPCN2 hypomorphic variants in the Hungarian population and showed that the patient also carries several of these variants. They conclude that these variants are not pathogenic in albinism.

Among these various results, one does not understand what is the main focus of the manuscript. Is it about analysing Hungarian patients with albinism? Is it to describe NM_139075.4:c.2060A>G; p.N687S and say that is a VUS and it should not be considered by other researchers? Is it to describe TPCN2 allele associations in the Hungarian population?

3) The authors say throughout the text that there are 22 albinism genes. Strictly speaking, there are only 20. The FHONDA gene may be added. But what is the 22nd?

4) References 3 and 4 are not adequate for the number of albinism genes.

5) In the 11 patients with an established molecular diagnosis of albinism, was parental segregation confirmed in order to assure that the 2 variants are in trans in each case ?

6) The NM_139075.4:c.2060A>G; p.N687S variant is rare, but there are 67 heterozygotes and 1 homozygote in GnomAD. This clearly disqualifies it, in the context of a dominant phenotype. In comparison the NM_139075.4:c.628C>T;p.Arg210Cys formerly described by Wang et al. only has 1 heterozygote, and 0 homozygote. 

7) line 137: The patient does not have ocular signs of albinism, which are central in the clinical definition of the disease. One therefore cannot say that he has albinism. He may have a differential diagnosis.

8) line 211: TPCN2 is so far not acknowledged as an albinism gene ; therefore the authors cannot claim that this gene should be included in the albinism gene panel.

9) line 48: talking about genetic backgroud here is not correct.

10) line 132: sentence is not correct: variants in the genes are not present, but the gene were present.

Comments on the Quality of English Language

English is overall good, but may be improved in some places.

Reviewer 2 Report

Comments and Suggestions for Authors

This is a very nice piece of work. I would suggest getting an English editor for the finer points as some of this is a little hard to read for example:

"Elucidate the genetic background of the Hungarian albinism cohort. Using albinism 22 panel with 22 genes, the genetic background of 11 out of 17 Hungarian patients were elucidated" (Abstract)

"Clinical consequences of the novel N687S variant is predicted to be pathogenic (“del- 85 eterious” by the SIFT algorithm, “probably damaging” by the PolyPhen-tool, “likely dis- 86 ease-causing” by the REVEL method, “damaging” by the MetaLR method and “medium” 87 by the Mutation Assessor analysis tools of the Ensemble Genome Browser, 88 http://www.ensembl.org)." (Results)

It would also be good to see if these cases had other melanogenesis pathway variants such as other OCA2 variants, OA variants, TYR etc. There is mounting evidence that mutations in different albinism or pigment pathway genes can add together to yield a phenotype so a discussion of this particular perhaps the TYR R402Q and S192Y common allele story and reporting these variants in these cases would strengthen the argument.

The references for gene panel yield in Albinism patients could also be extended as I believe only one such study is referenced, but there are many others with various nuances in which genes were tested how much phenotyping was done etc.  This should be expaneded.

Comments on the Quality of English Language

As above. Some English revision would make this much easier to read.

Round 2

Reviewer 1 Report

Comments and Suggestions for Authors

Thank you for your answers and improvements to the manuscript. The functional data aiming to evaluate the effect of the TPCN2 variant are appreciated. 

My overall opinion remains that a manuscript reporting as a main and central point a benign variant is of extremely weak interest to the reader.

I understand very well that rare and common hypomorphic variants may have an impact on the phenotype but here clearly there is no effect at all with this variant. One can always say that the possible implication of modifying variants must be taken into account, this is absolutely right, but the data here do not support this possibility.

In total, the study is fine, the experimental data are sound, and they enable to conclude that this variant has no functionnal effect. This is a negative result and this is fine. But my opinion is that publishing this type of negative result is not justified.